# Design of Artificial Peptide Against HIV-1 Based on the Heptad-Repeat Rules and Membrane-Anchor Strategies

**DOI:** 10.3390/ph18121881

**Published:** 2025-12-12

**Authors:** Jiali Zhao, Yan Zhao, Xiao Qi, Xiaojie Lv, Yanbai Tang, Wei Zhang, Qingge Dai, Jiaqi Xu, Dongmin Zhao, Qilu Yan, Guodong Liang, Jianping Chen

**Affiliations:** 1Key Laboratory for Candidate medicine Design and Screening Based on Chemical Biology, College of Pharmacy, Inner Mongolia Medical University, Hohhot 010110, China; 13947981607@163.com (J.Z.); zhaoyan@immu.edu.cn (Y.Z.); 20210095@immu.edu.cn (X.Q.); lxj0471@126.com (X.L.); zhangwei0488@163.com (W.Z.); nydaiqingge@163.com (Q.D.); 15648616289@163.com (J.X.); 20250014@immu.edu.cn (D.Z.); 15124832348@163.com (Q.Y.); 2State Key Laboratory of Natural and Biomimetic Drugs, Peking University, Beijing 100191, China; 3Peptide Drugs Research and Development Center, Zhen-Xiang Technology Co., Ltd., Hohhot 011500, China

**Keywords:** HIV-1, heptad-repeat rules, membrane-anchor strategies, fusion inhibitors, artificial peptide

## Abstract

**Objective:** The six-helix bundle (6-HB) is critical for HIV-1 membrane fusion. To disrupt this process, peptide inhibitors have been meticulously designed to target interactions within the 6-HB regions, thereby blocking membrane fusion and exerting inhibitory effects. Current peptide inhibitors like Enfuvirtide suffer from drug resistance and short in vivo half-life. This study aims to design novel anti-HIV-1 peptides by integrating heptad-repeat rules and membrane-anchor strategies. **Methods:** Artificial peptides were designed using HR rules from the HIV-1 gp41 6-HB motif and membrane-anchor modifications. **Results:** EK35S-Palm has emerged as a highly promising candidate for HIV-1 inhibition, exhibiting robust binding affinity to the target and effectively impeding the 6-HB spontaneous formation. **Discussion:** HR-based design avoids viral sequence homology, and membrane anchoring enhances local agent concentration, improving pharmacokinetics. The HR binding and membrane stabilization of EK35S-Palm provide synergistic inhibition. **Conclusions:** Integrating HR structural design with membrane-anchor strategies yields potent HIV-1 fusion inhibitors. EK35S-Palm demonstrates superior efficacy and stability over current therapies. These approaches hold great potential for overcoming the current therapy limitations and advancing the more effective and durable HIV-1 fusion inhibitors.

## 1. Introduction

Human immunodeficiency virus (HIV) is the causative agent of acquired immune deficiency syndrome (AIDS), a severe and highly contagious disease marked by a substantial mortality rate. According to the 2023 report by the Joint United Nations Programme on HIV and AIDS (UNAIDS), in the neighborhood of 1.3 million individuals were newly infected with HIV, and approximately 39.9 million people and 630,000 deaths from AIDS-related illnesses (https://www.unaids.org, accessed on 1 January 2025). Traditionally, vaccination has proven to be the most potent strategy in preventing infectious diseases. However, despite extensive and relentless research endeavors, the scientific community has yet to develop a safe and efficacious vaccine against HIV [1]. Consequently, the reliance on pharmacological interventions, particularly highly active anti-retroviral therapy (HAART), stands as the primary strategy for the management of AIDS [2]. While significant strides have been made in improving treatment options, the ongoing quest for a definitive therapeutic solution remains an urgent and critical priority.

Currently, HIV therapeutics approved by FDA encompass reverse transcriptase inhibitors (RTIs), protease inhibitors (PIs), integrase inhibitors, capsid inhibitors (e.g., lenacapavir), and entry inhibitors. [3]. One kind of entry inhibitor known as a fusion inhibitor (FI) functions by preventing the early stage for infection fusion. As the first peptide fusion inhibitor, Enfuvirtide (T20) received FDA approval in 2003 to treat HIV-1 infections [4]. In recent years, Sifuvirtide received approval for clinical phase III trials, and Albuvirtide received approval in China [5]. It implies that the use of peptide-based fusion inhibitors in HIV-1 treatment is becoming increasingly popular.

The membrane fusion process of HIV-1 with its host cell, a key step in viral entry mediated by the envelope glycoprotein, has been meticulously characterized (Figure 1). The envelope glycoprotein (Env), which consists of the transmembrane glycoprotein gp41 and the surface glycoprotein gp120, plays a central role in mediating HIV-1 infection of target cells. Within gp41, the N-terminal heptad repeat (NHR) and C-terminal heptad repeat (CHR) regions play critical roles [6]. Initially, gp120 binds to the CD4 receptor on host cell membrane, followed by interaction with either the CXCR4 or CCR5 co-receptors. This triggers the gp41 region exposure, previously concealed within gp120, and initiates a cascade of conformational changes. The NHR forms a coiled-coil core through interactions among three molecules, known as the N-Trimer, which then facilitates the anti-parallel binding of three CHR molecules to its hydrophobic grooves, resulting in the formation of the six-helix bundle (6-HB) [7]. N36 is a well-characterized peptide fragment corresponding to the NHR region of gp41 (residues 546–581), while C34 is a peptide derived from the CHR region (residues 628–661). These two peptides spontaneously associate in vitro to form a stable 6-HB structure, serving as a canonical model for studying Env-mediated fusion and evaluating the activity of fusion inhibitors. The fusion pore hinges on this structural transition, which bridges the viral envelope with the host cell membrane and ultimately facilitates membrane fusion. Consequently, the 6-HB represents not only a pivotal therapeutic target but also the primary molecular source for designing HIV-1 peptide-based fusion inhibitors. By targeting the 6-HB intermediate or directly disrupting 6-HB formation, peptide-based fusion inhibitors can effectively block HIV-1 entry into host cells, offering a promising therapeutic strategy against HIV-1 infection [8].

It is worth noting that, within 6-HB structural arrangement, the CHR domains are partitioned into two distinct regions: a buried hydrophobic interface involved in binding and exposed hydrophilic regions accessible to solvent [9]. Structural studies have revealed a remarkable tolerance for sequence variability in the CHR motifs of the HIV-1 6-HB, provided the amphiphilic nature of the helices driving the coiled-coil assembly remains intact. Through the implementation of approaches such as salt bridges, helix-favoring amino acids, and hydrophobic mutations in buried residues, researchers have developed α-helix-constrained C-peptides [8,10]. These peptides exhibit significantly enhanced bundle stability compared to their parent CHR regions, even though nearly 50% of the native residues were replaced. Furthermore, incorporating membrane-anchored small molecules into peptide tails is a further method to enhance the antiviral ability of peptide inhibitors. Recent research has demonstrated that lipid-conjugated C-peptides exhibit significantly increased α-helicity and enhanced binding capability to NHR region [8].

Capitalizing on the modifiable nature of the 6-HB α-helical structure and the design advantages of artificial peptides, we engineered novel artificial peptides by replicating the CHR helix topology through integration of custom-engineered amphipathic α-helical motifs and a lipid-targeting membrane anchor. As a result, the top-performing compound, EK35S-Palm, effectively targets the NHR region to prevent endogenous 6-HB formation, exhibiting robust antiviral efficacy against HIV-1, while demonstrating better metabolic stability than T20. This artificial-peptide design strategy holds significant promise for developing novel antivirals targeting the HIV-1 membrane fusion process.

## 2. Design

In the heptad-repeat region, understanding molecular interactions—such as hydrophobic, electrostatic, and hydrogen bonding—is critical, as these non-covalent forces drive the 6-HB formation [11]. Artificial peptides can be rationally designed according to the CHR domain distribution rules, exhibiting excellent anti-HIV-1 activity by binding to the NHR domain through non-covalent interactions. Building on this, Shi et al. [12] introduced partial natural CHR sequences at the 5HR, viz. (AEELAKK)_5_, N-terminus and mutated (*a*,*d*) positions within the heptad-repeat rules, resulting in the artificial peptide PBD-m4HR, which exhibited comparable activity to natural C-peptides. This suggests that rule-based mutagenesis introduces key residues that enable the α-helical artificial peptides, which have no homology to the HIV-1 envelope protein, to mimic the CHR active conformation and enhance binding with the NHR region to exert effective anti-HIV-1 activity. Thus, the heptad-repeat rules provide crucial insights for our first round in designing artificial peptides.

Covalent bonding of lipid molecules such as fatty acids, cholesterol, and pentacyclic triterpenoids on peptides enhances the lipophilicity, which can be anchored to “lipid rafts” mediating the viral membrane fusion, effectively increasing the local drug concentration near the target region [13,14]. Research demonstrates that multiple peptide-conjugated fusion inhibitors, such as C34-conjugated cholesterol, DP20-conjugated fatty acids, and P26-conjugated pentacyclic tricyclic compounds, exhibit not only significantly enhanced anti-HIV-1 activity but also improved pharmacokinetic properties [13,15,16]. For de novo-designed amphiphilic α-helical peptides, the introduction of membrane-anchored small molecules can further stabilize the helical active conformation and enhance the antiviral activity by anchoring to the membrane fusion region. Therefore, the membrane-anchor strategies are the core tactic in our second round of optimizing artificial peptides.

Based on the heptad-repeat rules, the first round of artificial-peptide design was conducted as follows: (i) Alanine (Ala) residues at the (*a*,*e*) positions in (AEELAKK)_n_ were mutated to Isoleucine (Ile) residues to provide hydrophobic interaction encapsulated in the NHR pocket domain [17]; (ii) the 6-HB crystal structure reveals that the CHR region hovering outside interacts with the NHR through key residues at the (*a*,*d*,*e*) positions, while the residues at the (*b*,*c*,*f*,*g*) positions are the non-target binding regions [18]. Therefore, we introduced double E-K salt bridges at the (*b*,*f*) and (*c*,*g*) positions to stabilize the artificial helix structure, immobilized hydrophobic amino acid residues at (*a*,*d*) positions, and introduced nonpolar, polar, and aromatic amino acid residues at (*e*) positions, respectively, in order to search for the strongest binding to the target (e.g., hydrogen bonding, electrostatic interactions, π-π stacking, etc.). Through the first design round, we proposed to obtain the most appropriate artificial-peptide sequence matching the target with potential anti-HIV-1 activity (Table 1).

For the second round of artificial-peptide design, in an attempt to improve the peptide membrane affinity to achieve targe enrichment effect, we chose oleanolic acid, cholesterol, and palmitic acid as the membrane-anchoring molecules to modify the above peptides (Figure 2): (i) Pep-OApc: the carboxyl group at the C3 position of oleanolic acid was protected by allyl group and the hydroxyl group at the C28 position was protected by pentynyl group, named OApc, and the azide group was strategically incorporated at the peptide N-terminus, the Pep-OApc was finally obtained via click chemistry; (ii) Pep-Chol: the Cys residue was introduced at the peptide C-terminus, and the hydroxyl group at the C3 position of cholesterol was condensed with bromoacetic acid to obtain cholesteryl bromoacetate. The bromine atom attacks the Cys sulfhydryl group of the peptide to obtain the Pep-Chol; (iii) Pep-palm: palmitic acid is able to condense with the peptide on the solid phase resin to obtain the Pep-palm. We intend to carry out HIV-1 inhibitory activity screening and mechanism exploration on the above obtained peptide–small-molecule conjugates, aiming to identify novel anti-HIV-1 lead compounds, and finally to establish a mature and effective methodology to expedite the development of innovative peptide-based fusion inhibitors.

## 3. Results

### 3.1. Artificial Peptides Demonstrated Potential Inhibition Activity Against HIV-1

We conducted activity screening of the artificial peptides using HIV-1 Env-mediated cell–cell fusion assay and neutralization assay of HIV-1 pseudovirus. For the five peptides designed in the first round, variation in the types of amino acid residues at the (*e*) position elicited marked disparities in their anti-HIV-1 efficacy profiles (Table 2). Peptides with polar amino acid residues generally exhibited stronger inhibitory activity compared to those with non-polar residues, with EK35S exhibiting the optimal performance against HIV-1 (cell–cell fusion inhibitory activity: 5.98 μM, HIV-1 pseudovirus inhibitory activity: 2.42 μM).

To further enhance anti-HIV-1 efficacy, we selected EK35S as the parent sequence, then covalently modified it with oleanolic acid, cholesterol, and palmitic acid to improve membrane affinity and achieve target site enrichment. Subsequent screening of the three peptide–small-molecule conjugates revealed that EK35S-Chol exhibited weak HIV-1 inhibitory activity, significantly lower than the parent EK35S. This reduction in activity may be attributed to the rigid structure of cholesterol disrupting the α-helical conformation of EK35S, as confirmed by circular dichroism (CD) spectroscopy, which showed a dramatic decrease in helical secondary structure for EK35S-Chol compared to EK35S (Appendix A). In contrast, EK35S-OApc displayed anti-HIV-1 ability comparable to the parent EK35S, suggesting that oleanolic acid may not effectively contribute to peptide anchoring or increase target site concentration. Surprisingly, EK35S-Palm demonstrated notable antiviral potency against HIV-1, with 13 nM cell–cell fusion inhibitory activity and 12 nM HIV-1 pseudovirus inhibitory activity, which is close to the positive control T20, and displayed over 200-fold more potent than the parent EK35S (Table 2 and Figure 3). Furthermore, the selectivity index (SI) of EK35S-Palm is >11,538 (SI = CC_50_/IC_50_ for inhibiting HIV-1 Env-mediated cell–cell fusion), suggesting that EK35S-Palm is a potent inhibitor of HIV-1. This significant enhancement indicates that palmitic acid modification could enrich the compound at the membrane fusion site, thereby improving antiviral efficacy. This approach aligns with the strategy used in the design of the broad-spectrum artificial peptide inhibitor IIQ for Middle East Respiratory Syndrome Coronavirus (MERS-CoV) and Influenza A Viruses (IAVs) [17].

### 3.2. Inhibition of Artificial Peptides Binding gp41 NHR Target on 6-HB Formation

We preferred EK35S and EK35S-palm to investigate the anti-HIV-1 mechanism. In CD experiments (Figure 4), the helical structure of EK35S-Palm was significantly more pronounced than that of the parent EK35S, indicating that the introduction of the palmitic acid (Palm) tail facilitates the formation of the helix-active conformation in the peptides. Upon mixing and incubating EK35S and EK35S-Palm with the target N36, the measured values of the blue curve and theoretical values of the green curve showed significant differences, confirming that the artificial peptides interact with the target N36 [19]. Comparable antiviral effects were observed when other artificial peptides were co-incubated with N36 (Appendix A).

Building on the previous experiments, N-PAGE and SE-HPLC were employed to investigate the interaction of artificial peptides with the target N36 in greater detail [18]. As shown in Figure 5, N36 showed no bands in lane 1, 5, 6, 7, 8 due to positive charge. In contrast, C34, which is negatively charged, displayed bands away from the starting point in lanes 2, 3, 6, 7, and 8. Lane 3 revealed the natural 6-HB band formed by N36 and C34. In lanes 6 to 8, as the concentration of EK35S and EK35S-Palm increased, the 6-HB bands in Figure 4B gradually became significantly lighter, while no obvious change was observed in the 6-HB band in Figure 4A. The retention time of peptides or complexes in SE-HPLC is correlated with their molecular size, with larger complexes exhibiting shorter retention times. As shown in Figure 5, the natural 6-HB complex formed by N36 mixed with C34 (blue line) had the largest molecular size and thus a significantly shorter retention time compared to C34 alone (red line). Upon the gradual addition of EK35S-Palm to the C34/N36 mixture, the chromatographic peaks corresponding to the 6-HB complex decreased significantly. Conversely, EK35S does not affect 6-HB peak (Figure 6). This observation aligns with the results from N-PAGE. In summary, the above experiments demonstrate that EK35S-Palm significantly disrupts native 6-HB compared to EK35S, thereby markedly enhancing anti-HIV-1 activity.

### 3.3. EK35S-Palm Binds to the Endogenous gp41 NHR Peptide to Form Polymer

CD analysis has demonstrated that EK35S-Palm can form a helical complex when mixed with the target N36. However, the precise binding forces between EK35S-Palm and N36, as well as the specific configuration of the resulting complex, remain unclear when analyzed solely through N-PAGE and SE-HPLC. To gain deeper insights, we employed settling velocity analysis (SVA) to investigate the properties of the EK35S-Palm/N36 mixture. The SVA results showed that the EK35S-palm/N36 mixtures contained two distinct molecular weights, 4.62 kDa and 22.1 kDa (Figure 7). The 4.62 kDa component corresponds to the molecular weight of EK35S-Palm, while the 22.1 kDa component closely matches the theoretical molecular weight of a hexamer formed by EK35S-Palm and N36, suggesting that EK35S-Palm can form a pseudo-6-HB with N36 [18]. These findings highlight the ability of EK35S-Palm to interact specifically and stably with the HIV-1 gp41 NHR region, ultimately forming multimeric bundles. These combination results (CD, N-PAGE, SE-HPLC and SVA) suggest that EK35S-Palm can interact with the gp41 NHR region to form stable multimers, which may provide a structural basis for interfering with HIV-1 membrane fusion.

### 3.4. The Binding of Artificial Peptides to the NHR Region Target Is Facilitated via Hydrogen Bonding and Electrostatic Interaction

Molecular docking provides a theoretical framework to validate the feasibility of artificial peptides binding to the NHR region target and to elucidate specific interactions. We utilized the internal NHR region from the 6-HB crystal structure (PDB ID: 1AIK) as the receptor and EK35S and EK35S-Palm as ligands for docking simulations using Schrödinger software (Schrödinger, New York, NY, USA; Version 2021-2).

In Figure 8A, the interaction schematic between EK35S and the target is depicted. Residues Glu9, Ser5, Glu2, and Glu30 of EK35S form hydrogen bonds with Arg557, Gln551, Gln550/Asn554, and Gln575/Arg579 in the NHR region, respectively. Additionally, Glu9 and Glu30 of EK35S engage in electrostatic interactions with Arg557 and Arg579 of the NHR region. All these interactions occur within 2 Å, and the calculated binding free energy (MM-GBSA) is −102.96 kcal/mol. In Figure 8B, EK35S-Palm demonstrates enhanced interactions with the NHR region. Residues Glu9, Ser12, Ser5, Glu30, and Glu2 of EK35S-Palm form hydrogen bonds with Arg557, Gln563, Gln551, Arg579, and Gln550/Asn554 of the NHR region, respectively. Furthermore, Glu9 and Glu30 of EK35S-Palm also exhibit electrostatic interactions with Arg557 and Arg579 of the NHR region, all within 2 Å. Notably, EK35S-Palm not only forms more interactions with the NHR region but also exhibits a higher binding free energy (MM-GBSA = −114.36 kcal/mol).

Previous studies have highlighted the critical role of hydrogen bonding and electrostatic interactions in the binding of fusion inhibitors to the NHR region [18]. Thus, in addition to the encapsulated hydrophobic interactions, EK35S and EK35S-Palm achieve favorable antiviral activity by tightly binding to the target through multiple hydrogen bonds and electrostatic interactions. This binding prevents the formation of the endogenous 6-HB, effectively inhibiting HIV-1 fusion and replication.

### 3.5. EK35S-Palm Exhibits Improved Phase I Metabolic Stability Compared to T20

Hepatic microsomes contain majority of phase I enzymes, with the microsomal mixed-function oxidase complex holding central significance among these systems. This enzymatic apparatus, characterized by cytochrome P450 isoforms as its catalytic core, forms the functional basis of xenobiotic metabolism, and the phase I metabolism system can be constructed in vitro by adding the corresponding cofactor NADPH. Based on this, we established the testing model for the phase I metabolic stability and investigated EK35S and EK35S-palm with in vitro warming-incubation method, leaving T20 as the control. As shown in Table 3, EK35S exhibited a slightly shorter Elimination half-life (Ehl) and a marginally higher Hepatic microsomal intrinsic clearance (Hmic) compared to T20, indicating that EK35S maintained a similar phase I metabolic stability profile to T20. Unexpectedly, EK35S-palm demonstrated a significantly longer Ehl of 216.56 min compared to T20, accompanied by an even lower Hmic. These findings underscore the superior phase I metabolic stability of EK35S-palm. Thus, not only have artificial peptides shown HIV-1 inhibition, but through palmitic acid modification, they also exhibit even more outstanding performance in phase I metabolism than the positive control T20.

## 4. Materials and Methods

### 4.1. Compounds Synthesis

All peptides, and EK35S-Palm, were synthesized using standard Fmoc solid-phase synthesis techniques. The synthesis was performed on Fmoc-protected rink amide resin with Fmoc-protected amino acids. Coupling reactions were carried out using O-benzotriazol-1-yl-N,N,N′,N′-tetramethyl-uronium hexafluorophosphate (HBTU, GL Biochem, Shanghai, China), 1-hydroxybenzotriazole (HOBt, GL Biochem, Shanghai, China), and diisopropylethylamine (DIEA, J&K Scientific, Beijing, China) in N,N-dimethylformamide (DMF, J&K Scientific, Beijing, China) solution. The Fmoc protective group was removed with 20% piperidine/DMF, and the resin was cleaved using a reagent mixture containing trifluoroacetic acid/m-cresol/thioanisole/water [5:0.2:0.2:0.1 (*v*/*v*/*v*/*v*)].

EK35S-OApc Synthesis [16]: Briefly, the purified azido-peptide precursor (1.0 equiv, 0.005 mmol) was dissolved in 2 mL of H_2_O, followed by the addition of triterpene derivatives (1.2 equiv, 0.006 mmol) dissolved in 1 mL of tert-butyl alcohol (J&K Scientific, Beijing, China). Subsequently, CuSO_4_·5H_2_O (J&K Scientific, Beijing, China) (1.0 equiv, 0.005 mmol) and sodium ascorbate (J&K Scientific, Beijing, China) (5.0 equiv, 0.025 mmol) were added to the mixture, which was stirred at room temperature for 4 h. The reaction progress was monitored by analytical RP-HPLC (Shimadzu analytical HPLC system, Shimadzu Corporation, Kyoto, Japan).

EK35S-Chol Synthesis [20]: Briefly, 20 mg of the purified peptide precursor, which included a Cys residue added to the C-terminus via a β-alanine linker, was dissolved in 300 μL of DMSO (J&K Scientific, Beijing, China). To this solution, 1.5 mg of cholest-5-en-3-yl bromoacetate (J&K Scientific, Beijing, China) dissolved in 200 μL of tetrahydrofuran (THF, J&K Scientific, Beijing, China) was added, followed by the addition of 7 μL of DIEA. The mixture was stirred at room temperature, and the reaction progress was monitored by analytical RP-HPLC.

All crude peptides were purified using preparative RP-HPLC (Shimadzu preparative HPLC system, Shimadzu Corporation, Kyoto, Japan), and the purity of the peptides was confirmed to be ≥95% by analytical RP-HPLC (Shimadzu analytical HPLC system, Shimadzu Corporation, Kyoto, Japan). The molecular weights of the compounds were characterized using Matrix-Assisted Laser Desorption Ionization-Time of Flight Mass Spectrometry (MALDI-TOF-MS, Autoflex III, Bruker Daltonics, Bremen, Germany).

### 4.2. HIV-1 Env-Mediated Cell–Cell Fusion Assay [18]

Briefly, the fluorescent reagent Calcein AM (abcam, Shanghai, China) was used to label H9/HIV-1 IIIB cells (obtained from the Inner Mongolia Autonomous Region New Drug Screening Engineering Research Center in Inner Mongolia Medical University, Hohhot, China). First, 2.5 μL of Calcein AM was used to stain 2 × 10^5^ H9/HIV-1 IIIB cells. After labeling at 37 °C for 30 min, the cells were washed twice with PBS (Thermo Fisher Scientific, Waltham, MA, USA) and resuspended in fresh RPMI 1640 (Sigma-Aldrich, St. Louis, MO, USA) medium supplemented with 10% FBS. Next, 50 μL of diluted peptides and 50 μL of a mixture containing 2 × 10^5^ H9/HIV-1 IIIB cells/mL were added to a 96-well cell culture plate and incubated at 37 °C for 30 min. Subsequently, 1 × 10^5^ MT-2 cells were added to the mixture and cultured at 37 °C for 2 h. The fused cells were quantified using an inverted fluorescence microscope. The IC_50_ values were calculated using CalcuSyn software (Version 2.0).

### 4.3. Neutralization Assay of HIV-1 Pseudovirus [21]

Briefly, U87 CD4^+^ CCR5^+^ cells (obtained from the Inner Mongolia Autonomous Region New Drug Screening Engineering Research Center in Inner Mongolia Medical University, Hohhot, China) (1 × 10^4^ cells/well) were plated in 96-well tissue culture plates and incubated at 37 °C in a humidified 5% CO_2_ atmosphere for 12 h to allow cell adherence. Serial dilutions of the test peptides were prepared, and each dilution was mixed with the HIV-1 pseudovirus derived from the JFRL macrophagic strain (subtype B). After incubating the virus-peptide mixtures at 37 °C for 30 min to allow interaction, the mixtures were transferred to the pre-seeded U87 CD4^+^ CCR5^+^ cells. At 12 h post-infection, the culture medium was refreshed to remove unbound viruses and excess peptides. Forty-eight hours post-infection, the cells were lysed using Cell Culture Lysis Reagent (Promega, Madison, WI, USA), and the luminescence intensity (reflecting luciferase activity) was quantified using the Luciferase Assay System (Promega, E4030) according to the manufacturer’s instructions.

### 4.4. Cytotoxicity Assays

Briefly, 100 μL of cell suspension (1 × 10^5^ cells/mL) was dispensed into individual wells of a 96-well culture plate, followed by incubation at 37 °C in a 5% CO_2_ atmosphere for 12 h. Subsequently, 5 μL of serially diluted peptide preparations was added to each test well. Meanwhile, a blank control group (without peptide) and a positive control group (supplemented with 5 μL of 10% Triton X-100) were set up, and all groups were cultured at 37 °C with 5% CO_2_ for 48 h. Afterward, 10 μL of Cell Counting Kit-8 solution (Yeasen, Shanghai, China)was added to each well, and the plate was further incubated for 2 h. Finally, the absorbance at 450 nm was measured using a microplate reader.

### 4.5. Circular Dichroism (CD) Spectroscopy Analysis

Peptides, as well as their equimolar mixtures, were prepared at a final concentration of 20 μM in PBS (10 mM, pH 7.4) and incubated at 37 °C for 30 min. The secondary structure of the peptides was analyzed using a circular dichroism (CD) spectrometer (J-1500, JASCO Corporation, Kyoto, Japan) with the following parameters: 1.0 nm bandwidth, 180–260 nm wavelength range, 0.1 nm resolution, 0.2 nm data pitch, and 200 nm/min scanning speed.

### 4.6. Methods for Native Polyacrylamide Gel Electrophoresis (N-PAGE)

Peptides, as well as their equimolar mixtures, were prepared at a final concentration of 100 μM in PBS (10 mM, pH 7.4) and incubated at 37 °C for 30 min. After incubation, the peptide solutions were mixed with methyl green loading buffer in a 1:1 ratio, and 20 μL of each sample was loaded into the wells of the gel. Gel electrophoresis was performed at room temperature, initially at a constant voltage of 90 V for 0.5 h, followed by 150 V for 2–3 h. The gels were then stained with Coomassie Brilliant Blue G250 (Bio-Rad, Shanghai, China), and images were captured using the ChampGel 6000 Imaging System (Sage Creation Ltd., Beijing, China).

### 4.7. Methods for Size-Exclusion High-Performance Liquid Chromatography (SE-HPLC)

EK35S and EK35S-Palm, as well as their equimolar mixtures, were prepared at a final concentration of 100 μM in PBS (10 mM, pH 7.4) and incubated at 37 °C for 30 min. The peptide or peptide mixture was then applied to a 300 mm × 7.8 mm Phenomenex BioSep-SEC-S2000 column, which was pre-equilibrated with PBS (pH 7.4). The elution was performed at a flow rate of 1.0 mL/min, and the fractions were monitored at 210 nm.

### 4.8. Methods for Sedimentation Velocity Analysis (SVA) [18]

An analytical ultracentrifuge (Beckman Coulter ProteomeLab XL-A, Brea, CA, USA) was used for the sedimentation velocity analysis experiments. In brief, EK35S-palm was incubated with N36 in PBS (10 mM, pH 7.4) at 37 °C for 30 min. All samples were prepared at a final concentration of 200μM and were initially scanned at 5000 rpm for 10 min to identify the appropriate wavelength for data collection. Data were collected at 60,000 rpm at a wavelength of 280 nm. Sedimentation coefficient distribution, c(s), and molecular mass distribution, c(M), were calculated using the SEDFIT program (Beckman Coulter ProteomeLab XL-A, Brea, CA, USA).

### 4.9. Molecular Docking [18]

Molecular docking was performed using Schrödinger molecular modeling software (Schrödinger, New York, NY, USA). The initial monomer of the HIV-1 gp41 core structure was obtained from the Protein Data Bank (PDB ID: 1AIK), and each NHR region of the 6-HB was extracted as the receptor, designated as the ABC chain. The main structures of the EK35S and EK35S-Palm peptides, serving as ligands, were constructed using the Build Biopolymer from Sequence module with the OPLS_4 force field. Molecular docking was carried out using the Protein-Protein Docking (Piper) module in Schrödinger (Version 2021-2). The resulting docking conformations were analyzed and visualized using PyMOL software (Version 2.0).

### 4.10. Phase I Metabolic Stability Assay

Experiments were carried out using the Phase I Metabolic Stability Research Kit (IPHASE, Suzhou, China). Solution A in the kit was pre-incubated at 37 °C for 5 min, and solution B was prepared from PBS (100 mM), the subject (or positive substrate), and liver microsomes. The above two were mixed and immediately placed in a 37 °C water bath for incubation. The incubation solution is quantitatively removed from the incubation system at preset time points (0, 5, 10, 15, 30, 60 min), and an equal volume of termination solution is added. Each sample was run in parallel three times. The substrate remaining amount at each time point was determined by RP-HPLC (Shimadzu preparative HPLC system, Shimadzu Corporation, Kyoto, Japan), the substrate remaining percentage was calculated, and the slope k was determined by linear regression analysis to calculate the Elimination half-life and Hepatic microsomal intrinsic clearance.

## 5. Conclusions

Inspired by the membrane fusion mechanisms observed in various type I enveloped viruses that typically form a six-helix bundle (6-HB), and taking into account the success of unnatural peptide fusion inhibitors against MERS-CoV and IAVs, we have embarked on the innovative design of a series of artificial-peptide fusion inhibitors specifically aimed at targeting HIV-1. Notably, these peptides exhibit no homology to the viral sequences. In our pursuit to significantly amplify their inhibitory efficacy against HIV-1, we employed strategies based on heptad-repeat rules along with membrane-anchor techniques. These strategies have proven effective in enhancing the interactions between the peptides and their targets, ultimately leading to a successful disruption of the HIV-1 fusion process. The implications of our findings not only confirm the potential of peptide-based inhibitors but also offer fresh perspectives for the advancement of novel HIV-1 fusion inhibitors, thereby contributing to the evolution of antiviral therapies in this domain.

## Figures and Tables

**Figure 1 pharmaceuticals-18-01881-f001:**
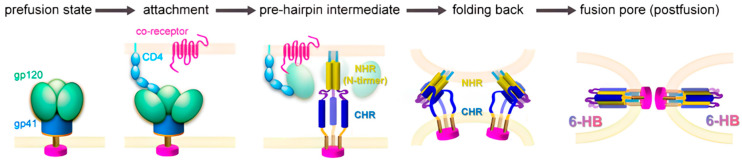
Schematic diagram of the HIV-1 membrane fusion (Adapted from J Virol. 2025; 99(5): e0228924 with permission) [3].

**Figure 2 pharmaceuticals-18-01881-f002:**
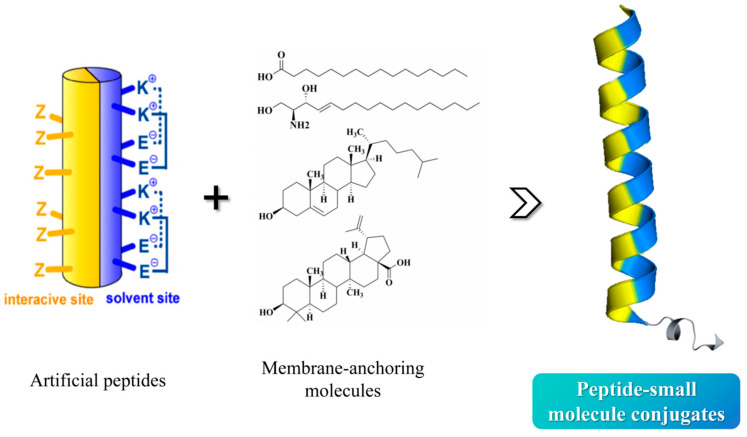
The peptide–small-molecule conjugates design schema.

**Figure 3 pharmaceuticals-18-01881-f003:**
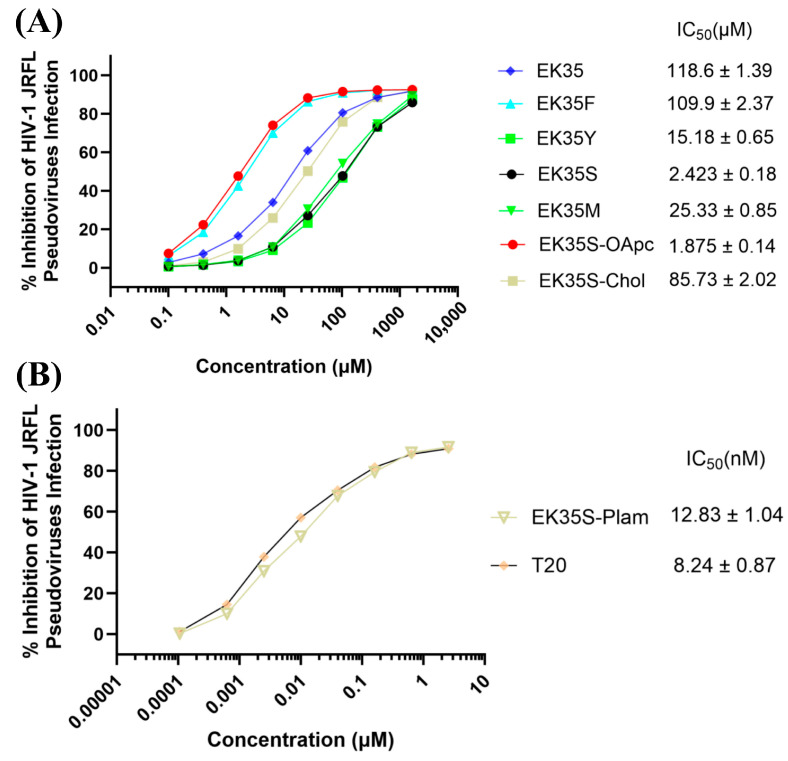
Inhibitory activity of artificial peptides in pseudovirus infection assays against HIV-1. (**A**) Inhibition by artificial peptides, except for EK35S-Plam. (**B**) Inhibition by EK35S-plam and T20.

**Figure 4 pharmaceuticals-18-01881-f004:**
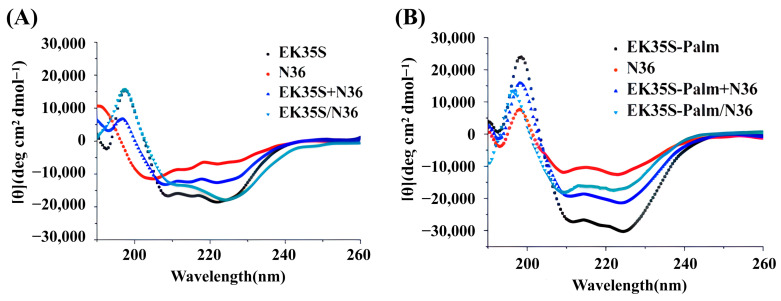
(**A**) The CD Spectrum of EK35S/N36. (**B**) The CD Spectrum of EK35S-Palm/N36.

**Figure 5 pharmaceuticals-18-01881-f005:**
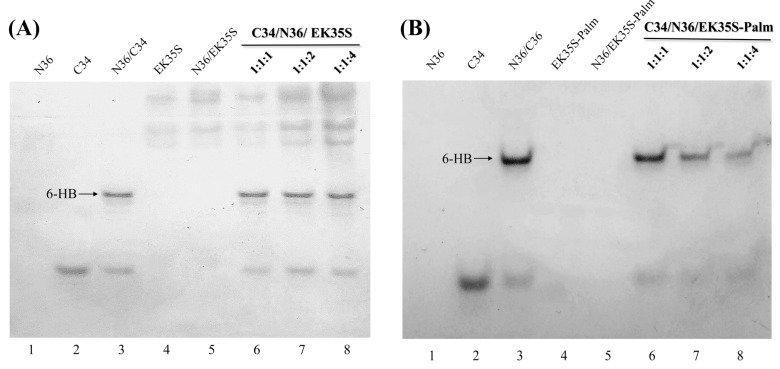
(**A**) N-PAGE to detect the blocking of 6-HB formation between C34 and N36 by EK35S peptide. N36 (20 μM) with or without EK35S was incubated at 37 °C for 30 min, followed by the addition of C34 (20 μM). The mixture was incubated at 37 °C for another 30 min before loading into the gel; (**B**) N-PAGE to detect the blocking of 6-HB formation between C34 and N36 by EK35S-Plam peptide. N36 (20 μM) with or without EK35S-Plam was incubated at 37 °C for 30 min, followed by the addition of C34 (20 μM). The mixture was incubated at 37 °C for another 30 min before loading into the gel.

**Figure 6 pharmaceuticals-18-01881-f006:**
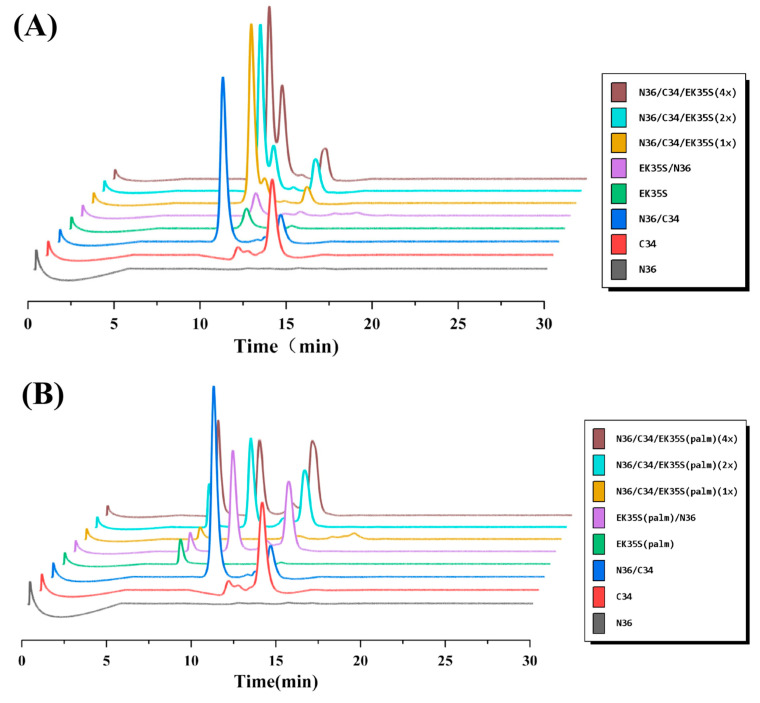
(**A**) SE-HPLC analysis of the interactions between the EK35S and N36/C34; (**B**) SE-HPLC analysis of the interactions between the EK35S-Palm and N36/C34.

**Figure 7 pharmaceuticals-18-01881-f007:**
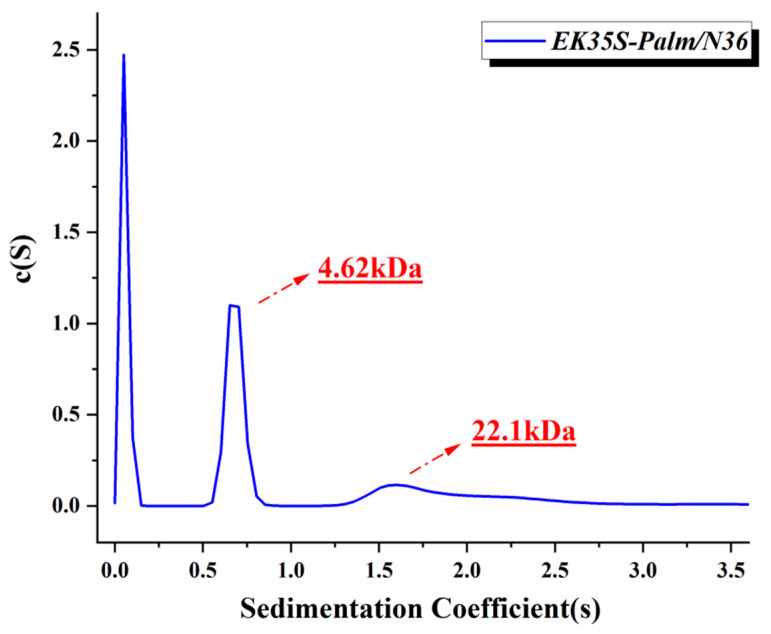
SVA result of the EK35S-Palm/N36 mixture.

**Figure 8 pharmaceuticals-18-01881-f008:**
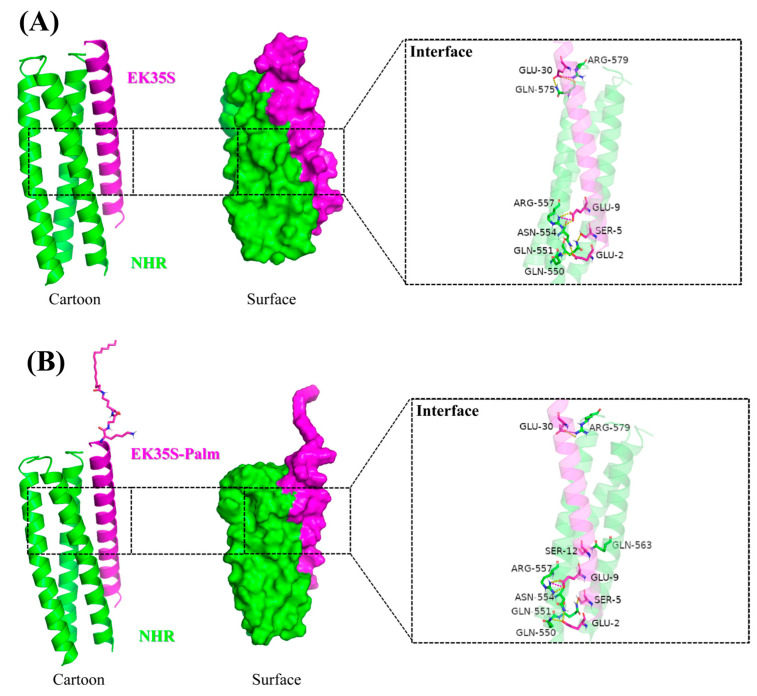
(**A**) Molecular docking map of EK35S with HIV-1 gp41 NHR region; (**B**) Molecular docking map of EK35S-Palm with HIV-1 gp41 NHR region.

**Table 1 pharmaceuticals-18-01881-t001:** The de novo-designed artificial peptides.

Name	Sequence
	abcdefg abcdefg abcdefg abcdefg abcdefg
EK35	IEELIKK IEELIKK IEELIKK IEELIKK IEELIKK
EK35F	IEELFKK IEELFKK IEELFKK IEELFKK IEELFKK
EK35Y	IEELYKK IEELYKK IEELYKK IEELYKK IEELYKK
EK35S	IEELSKK IEELSKK IEELSKK IEELSKK IEELSKK
EK35M	IEELMKK IEELMKK IEELMKK IEELMKK IEELMKK

Blue-highlighted residues represent amino acid mutation sites in the peptide sequence.

**Table 2 pharmaceuticals-18-01881-t002:** Inhibitory activity of artificial peptides on HIV-1 Env-mediated cell–cell fusion ^a^.

Peptide ^b^	Sequence	IC_50_ (μM)
EK35	IEELIKK IEELIKK IEELIKK IEELIKK IEELIKK	>100
EK35F	IEELFKK IEELFKK IEELFKK IEELFKK IEELFKK	92.36 ± 9.22
EK35Y	IEELYKK IEELYKK IEELYKK IEELYKK IEELYKK	23.15 ± 6.72
EK35S	IEELSKK IEELSKK IEELSKK IEELSKK IEELSKK	5.98 ± 1.23
EK35M	IEELMKK IEELMKK IEELMKK IEELMKK IEELMKK	15.76 ± 8.92
EK35S-OApc	IEELSKK IEELSKK IEELSKK IEELSKK IEELSKK-OApc	4.82 ± 0.89
EK35S-Chol	IEELSKK IEELSKK IEELSKK IEELSKK IEELSKK-Chol	>100
EK35S-Palm	IEELSKK IEELSKK IEELSKK IEELSKK IEELSKK-palm	0.013 ± 0.004
T20	YTSLIHSLIEESQNQQEKNEQELLELDKWASLWNWF	0.008 ± 0.002

^[a]^ Peptides were tested in triplicate, and the data are presented as the mean ± standard deviation. ^[b]^ All peptides have been N-terminally acetylated and C-terminally amidated.

**Table 3 pharmaceuticals-18-01881-t003:** Elimination half-life and Hepatic microsomal intrinsic clearance for EK35S, EK35S-Palm, T20 ^a^.

Compound	Ehl (Min) ^b^	Hmic (%) ^c^
EK35S	133.27	1.04
EK35S-Palm	216.56	0.64
T20	150.65	0.92

^[a]^ The chromatographic peak area was used as the calculation data. Take the zero concentration of the test substance as 100%, and compare the concentration at each time point with the zero concentration to obtain the remaining percentage. The natural logarithm of the remaining percentage of substrate at each time point is linearly regressed with the incubation time to obtain the slope *K*. ^[b]^ Elimination half-life = 0.693/*K.*
^[c]^ Hepatic microsomal intrinsic clearance = [0.693/Elimination half-life] × [Incubation volume (mL)/Liver microsomal mass (mg)].

## Data Availability

The original contributions presented in this study are included in the article/Appendix A. Further inquiries can be directed to the corresponding author(s).

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
