# Peer review of "Design of Artificial Peptide Against HIV-1 Based on the Heptad-Repeat Rules and Membrane-Anchor Strategies"

_pharmaceuticals, 2025, doi:10.3390/ph18121881_

Round 1
Reviewer 1 Report
Comments and Suggestions for Authors
The Authors describe the design of artificial peptides containing membrane-anchoring molecules as HIV-1 inhibitors targeting membrane fusion. The manuscript encloses the description of the peptides, their characterization and have been tested for their ability to interfere with HIV-1-Env-mediated cell-cell fusion.
Specific points:
- In the introduction section, among the HIV therapeutics approved by FDA should be mention also capsid inhibitor (lenacapavir indeed has been included in the approved drugs)
- In the introduction section, the Figure 1 is overlapping with part of the text ..as far as in view of this reviewer ..
- In the Design section, first paragraph...close to the end ... " ... with the NHR region to exert effective anti-HIV....I guess here "activity" is missing.
- In the Results section, no data about the effect of peptide on cell viability is reported. Regarding the biological assay on Env-mediated cell-cell-fusion, this reviewer has some concerns. The assay, as described in the Mat & Meth section, is not the standard one to test cell-cell fusion, at least to the knowledge of this reviewer. A reference is needed to support the reliability of the assay in this context. Otherwise, a different biological assay should be performed.
- Figure 3....how comes the spectrum for N36 by itself is different in the A and B panel ? shouldn't be the same ? please specify if experimental conditions are differente between the two panels, otherwise explain why and what could account for it. The same issue applies in Figure S14. Moreover, in Figure 3, would it be possible to provide percentage differences among different spectra ?
- Figure 4. According to this reviewer some overlapping exists between panel A and panel B, so the details are not visibile. Moreover, additional information should be provided in the legend to the figure to help the reader.
- Paragraph: 4.3...last sentence reports on the ability of "These results suggest that EK35S-Palm can interact with gp41 NHR region to form stable multiblundes...thus effectively inhibits HIV-replication". The authors do not report any data on inhibition of HIV-1 replication, thus inhibition of replication is not supported by the reported data and cannot be mentioned, as it is, as far as the point of view of this reviewer.
- Figure 1 is identical to the one reported in Huang et al. J Virol 2025, 99 (5). I don't know if this is allowed even though authors might appear in both manuscripts.... I think this should be specified (From....with permission)
Minor points:
- References are not listed as requested by the journal (for instance the year of publication should be in bold....). Moreover, reference #9 is incomplete and actually should be reported as 2025 not 2024 ...reference #1: the year 2024 appeared twice.
- min or minutes ? it should be uniform throughout the text
- the section regarding "Author contribution" is missing.
Reviewer 2 Report
Comments and Suggestions for Authors
The present study by Zhao et al. focuses on the design of new peptide inhibitors that block HIV-1 membrane fusion by targeting the six-helix bundle (6-HB) structure of the viral gp41 protein. Current inhibitors, such as Enfuvirtide, are limited by resistance and short half-life. Using heptad-repeat (HR) rules and membrane-anchoring modifications, researchers developed artificial peptides with improved stability and potency. Among these, EK35S-Palm showed strong binding to gp41, effectively preventing 6-HB formation and demonstrating enhanced pharmacokinetic properties. Overall, the manuscript is well-written, and is suitable for publication in Pharmaceuticals. However, there are some minor points that, if addressed, will further improve the clarity and quality of the manuscript:
Comments
- Page 2: “…with its host cell has been meticulously characterized (Figure 1).” This sentence seems incomplete.
- The authors should provide a more detailed explanation of the N36 and C34 peptides.
- Page 7: Define the meaning of the acronyms MERS-CoV and IAVs.
- Legend figure 3(B): The CD spectrum of EK35S-palm/36.
- Page 11: “This binding prevents the formation of the endogenous 6-HB, effectively inhibiting HIV-1 fusion and replication.” Formally, this study demonstrates the ability of EK35S and EK35S-Palm to inhibit Env-mediated cell-cell fusion. Since the Env protein mediates both virus-cell and cell-cell fusion, a peptide that inhibits cell-cell fusion may interfere with viral entry as well. However, the ability to inhibit HIV replication must be verified experimentally, for example, by Env-pseudovirus infection assays. The manuscript would benefit from experimental data on this.
Round 2
Reviewer 1 Report
Comments and Suggestions for Authors
-Peptides cytotoxicity assays should be included (for both cell lines adopted in the two biological assays). The question of this Reviewer is…"Which kind of test was performed to analyze peptides cytotoxicity ?"
Minor revisions
-In the "MATERIALS AND METHODS" section:
3.3: …with HIV-1 pseudoviruses representing different subtypes.
Actually, acoording to this Reviewer, the Authors report only HIV-1 pseudotyped by the JFRL macroge-tropic strain …thus, it cannot be reported as “different subtypes”…what does this mean ?
-does not affect 6-HB peak conversely(Figure 6). : includes space before left bracket
-In the FUNDING section: includes space before the left bracket
….Autonomous Region of China(No.2025KJHZ0064)
…. Inner Mongolia Medical University(YKD2025BSQD023)
Round 3
Reviewer 1 Report
Comments and Suggestions for Authors
The manuscript including the last modifications according to this Reviewer is now suitable for publication